# Plasmonic computing of spatial differentiation

Tengfeng Zhu[1], Yihan Zhou[1], Yijie Lou[1], Hui Ye[2], Min Qiu[2], Zhichao Ruan[1,2] & Shanhui Fan[3]

Optical analog computing offers high-throughput low-power-consumption operation for specialized computational tasks. Traditionally, optical analog computing in the spatial domain uses a bulky system of lenses and filters. Recent developments in metamaterials enable the miniaturization of such computing elements down to a subwavelength scale. However, the required metamaterial consists of a complex array of meta-atoms, and direct demonstration of image processing is challenging. Here, we show that the interference effects associated with surface plasmon excitations at a single metal–dielectric interface can perform spatial differentiation. And we experimentally demonstrate edge detection of an image without any Fourier lens. This work points to a simple yet powerful mechanism for optical analog computing at the nanoscale.

[1] State Key Laboratory of Modern Optical Instrumentation, Department of Physics, Zhejiang University, Hangzhou 310027, China. [2] State Key Laboratory of Modern Optical Instrumentation, College of Optical Engineering, Zhejiang University, Hangzhou 310027, China. [3] Department of Electrical Engineering, Ginzton Laboratory, Stanford University, Stanford, California 94305, USA. Correspondence and requests for materials should be addressed to Z.R. (email: zhichao@zju.edu.cn) or to S.F. (email: shanhui@stanford.edu).

Differentiation is a fundamental mathematical operation used in any field of science or engineering. In image processing, spatial differentiation enables edge detection, which extracts important information about the boundary of objects in an image[1,2]. Edge detection is the essential first step in object detection, feature classification and data compression[3]. However, in many applications that require real-time processing of images, such as in medical and satellite applications, essential high-throughput edge detection demands time-consuming computation and represents a key challenge[4,5].

Over the past few years, optical analog computing has attracted particular attentions and offers high-throughput low-power-consumption operation for specialized computational tasks. In the temporal domain, such computing allows direct and coherent transformation of pulse waveforms that are too short to be processed directly electronically for applications from analog computing[6–13] and optical memory[14] to differential equation solvers[15,16] and photonic neural networks[17]. In the spatial domain, such computing enables massively parallel processing of entire images with no energy cost, which provides significant advantages against standard digital processing of images. Traditionally, optical analog computing in the spatial domain uses a bulky system of lenses and filters[18]. Recently, there are significant efforts seeking to miniaturize such computing elements down to a single wavelength or even subwavelength scale[19–29]. In particular, Silva et al.[19] theoretically showed that through a complex array of meta-atoms both phase and magnitude of the transmitted wave can be engineered to realize desired spatial transfer functions of mathematical operations such as spatial differentiation. However, due to the structure complexity of the metamaterials, direct experimental demonstration of image processing is challenging[21,24]. There are also several other theoretical proposals, including one discussed by Silva et al.[19], where one seeks instead to achieve spatial differentiation with layer structures[26–28]. None of these structures, however, have been demonstrated experimentally neither.

Here, we experimentally demonstrate an optical spatial differentiator based on the simplest surface plasmonic structure, a single layer of thin metal film, and without any Fourier lens. In contrast to the metamaterial approach for spatial differentiation, which relies on a complex array of meta-atoms, the proposed plasmonic differentiator consists of a single ultra-thin metal film[29], which is far simpler and greatly simplifies the fabrication process and reduces the influence of fabrication imperfection. Moreover, such a plasmonic structure enables the differentiator to have the thickness of the film reduced into the subwavelength scale about 50 nm, which represents a tremendous miniaturization as compared to lens-based Fourier optical systems or even the dielectric slab approaches. Also we exploit such a differentiator to detect the edges for sharp changes in either amplitude or phase of the incident field. With standard thin-film deposition technology, the plasmonic differentiator can be fabricated on a large scale. Therefore, we have demonstrated an ultrafast optical computing scheme with sufficient space-bandwidth product that is capable of processing an entire image on a single shot, which is critically important for high-throughput real-time image processing.

## Results

**Surface-plasmon-based scheme for spatial differentiation.**
To achieve spatial differentiation, we consider the Kretschmann prism configuration as schematically shown in Fig. 1a. Here, we consider that a p-polarized wave obliquely illuminates the metal surface from the glass side with an incident angle $\theta_0$. Suppose that

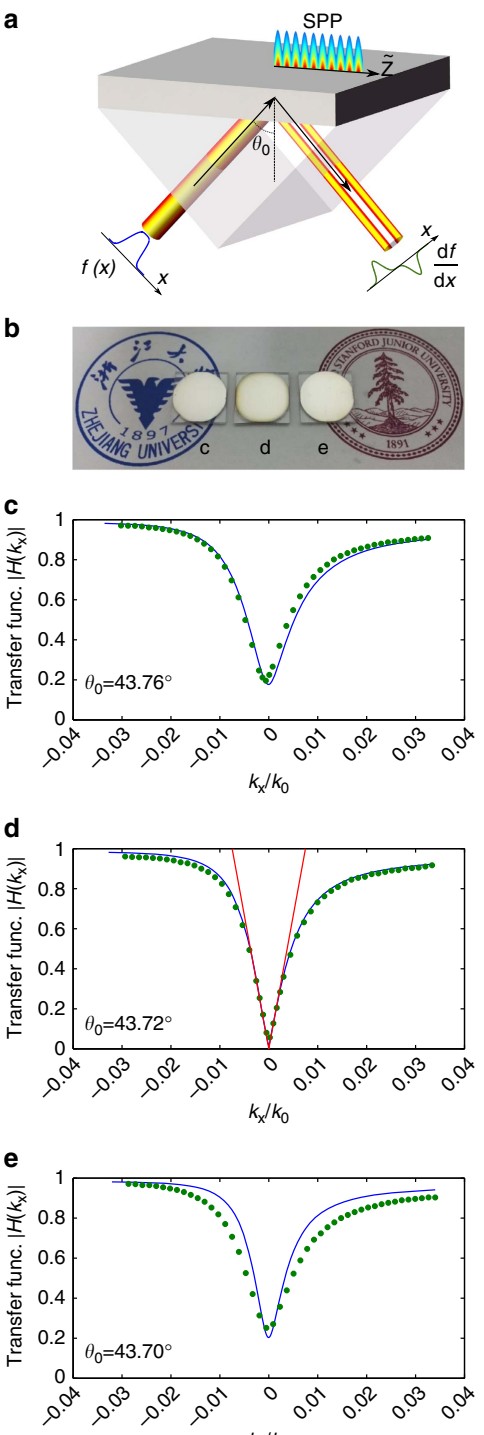

**Figure 1 | Design of surface-plasmon-based spatial differentiator.**
(**a**) Schematic of the plasmonic spatial differentiator with the Kretschmann configuration to excite the surface plasmon polariton (SPP). The dark grey layer and the light grey area correspond to the silver film and the glass, respectively. (**b**) Sample photo of three deposited silver thin films with different thicknesses. (**c**–**e**) Spatial transfer function spectra of the three samples by the experimental measurement (dotted lines) and the numerical fitting (solid lines). In the numerical calculation, the thicknesses of the silver layer were 46.0, 50.0 and 55.5 nm, respectively. $\theta_0$ corresponds to the incident angle for the phase matching. The red line in (**d**) corresponds to the fitting results to equation (2).

the incident and the reflected beams have the field profiles $S_{in}(x)$ and $S_{out}(x)$, respectively, where $x$ is the beam profile coordinate which is perpendicular to the beam propagation direction and the magnetic field. By spatial Fourier transform, the incident (reflected) beam can be written as the superposition of plane waves, that is $S_{in(out)}(x) = \int s_{in(out)}(k_x)\exp(ik_x x)dk_x$, where $k_x$ is the wavevector component of the plane wave along $x$ direction, and $s_{in(out)}(k_x)$ are the corresponding amplitudes. Therefore, the field-profile transformation from the incident to the reflected light is described by a spatial spectral transfer function $H(k_x)$ where $H(k_x) \equiv s_{out}(k_x)/s_{in}(k_x)$.

In the special case when the incident angle $\theta_0$ satisfies the phase matching condition, that is when the plane wave has a wavevector component parallel to the interface that matches with the wavevector of a propagating surface plasmon polariton (SPP) at the metal–air interface, the incident plane wave then strongly excites the SPP[30,31]. Due to material loss, the decaying SPP results in a dip of the transfer function spectrum $H(k_x)$ at $k_x = 0$. Furthermore, as the excited SPP propagates along the metal surface it leaks out and radiates. Therefore, the reflected amplitude is determined through the interference between the direct reflection at the glass–metal interface and the radiation from the SPP leakage. Using the spatial coupled mode theory formalism[32,33], which accounts for the general behaviour of this system using the constraints of energy conservation, time-reversal symmetry and mirror symmetry, the spatial spectral transfer function around $k_x = 0$ can be expressed as

$$H(k_x) = e^{i\varphi}\frac{ik_x + A}{ik_x + B},\tag{1}$$

where $\varphi$ corresponds to the phase change during the direct reflection at the glass–metal interface. $A = (\alpha_{spp} - \alpha_l)/\cos\theta_0$ and $B = (\alpha_{spp} + \alpha_l)/\cos\theta_0$, where $\alpha_l$ is the radiative leakage rate of the SPP, and $\alpha_{spp}$ is the intrinsic material loss rate (for the details of derivation see Supplementary Note 1 and Supplementary Fig. 1).

We note that $A = 0$ when the critical coupling condition $\alpha_l = \alpha_{spp}$ is satisfied. Near $k_x = 0$ equation (1) can then be approximated as

$$H(k_x) \approx \frac{e^{i\varphi}}{B}ik_x.\tag{2}$$

Equation (2) is the transfer function of a first-order spatial differentiator. Correspondingly, in the spatial domain, the reflected field profile becomes

$$S_{out}(x) = \frac{e^{i\varphi}}{B}\frac{dS_{in}}{dx}.\tag{3}$$

Equation (3) shows that spatial differentiation can be realized by exploiting the interference effect associated with the surface plasmon excitation without the need of any lens that performs Fourier transform.

As a few remarks, we note that here we are able to demonstrate the first-order derivative since the plasmonic differentiator operates away from the normal incidence. This is in contrast to ref. 19, where with a multilayer film a second-order derivative is demonstrated for normal incidence due to the symmetry constraint on both the flat structure and the light source. Also, with a relatively low quality factor of the plasmonic resonance, when maintaining the incident angle, the reflection remains low with a terahertz bandwidth at the resonant frequency. Therefore, the spatial differentiator proposed here should have enough frequency bandwidth for ultrafast image processing.

**Design of SPP-based differentiator.** To experimentally realize the spatial differentiation, in the Kretschmann configuration as shown in Fig. 1a, we control the thickness of the metal film to satisfy the critical coupling condition. We note that the leakage rate $\alpha_l$ of the SPP at the metal–air interface monotonically decreases to zero as the metal film thickness increases to infinity. On the other hand, the intrinsic material loss rate $\alpha_{spp}$ is mainly determined by the metal loss and thus is insensitive to the thickness. Therefore, the critical coupling condition can be realized with an appropriate choice of the metal film thickness. By the transfer matrix method[34] we determined that the critical coupling condition is satisfied with the silver film thickness around 50 nm for an incident laser beam with a wavelength of 532 nm.

We deposited silver on a BK7 glass substrate (see the sample photo in Fig. 1b and the detailed information in the Methods) and measured the reflectance with the Kretschmann configuration in order to obtain the spatial transfer function spectrum. Figure 1c–e show the experimental spatial transfer function spectra of three samples with different thicknesses. All three films show a pronounced dip in the transfer spectrum that arises from the phase-matched coupling to the SPP on the metal–air interface. By fitting the experimental data with the numerical calculation, the thicknesses of the silver layers were estimated as 46.0, 50.0 and 55.5 nm. Here the dielectric constant of the silver is $\varepsilon_{Ag} = -11.51 + 0.55i$, which is close to the literature values[35,36] with a small deviation that is reasonable, given the depositing methods and conditions.

For the film with the thickness of 50.0 nm, the lowest value of the magnitude of the transfer function is as low as 0.0170, indicating that the structure is close to the critical coupling condition. Moreover, in consistency with the theoretical prediction above, Fig. 1d shows that the spatial spectral transfer function indeed exhibits linear dependency on $k_x$ around $k_x = 0$. After fitting with equation (2), we determine the parameter $B = 0.0075k_0$, where $k_0$ is the wavevector in air. We note that the relatively small value of $B$ enhances the output signal obtained from the spatial differentiation (see equation (3)). On the other hand, such a small value also means that the linear dependency is preserved only near $k_x = 0$ within a narrow spatial bandwidth (Fig. 1d). For the rest of the paper we use this sample for the demonstration of spatial differentiation and edge detection.

**Experimental demonstration.** As the first example of spatial differentiation we consider an incident field generated by sending a beam through a slit with an adjustable width. The incident and reflected beam intensities are shown in Fig. 2a,b, respectively. The reflected beam indeed shows sharp peaks located at the edges of the incident beam, providing a direct demonstration of spatial differentiation and edge detection.

As a quantitative illustration of the performance of spatial differentiation, we input the beam profiles with different slit widths as illustrated in Fig. 2c, and in Fig. 2d compare the resulting experimental reflected beam profiles with the results from numerical differentiation of the input beam profiles. For ease of presentation, for each $x$-value we perform an average along the $y$-direction. The solid lines in Fig. 2c,d correspond to the averaged intensities of the experimental incident and reflected ones after normalization. In Fig. 2c, the input beams correspond to slit widths 321.65, 504.20 and 738.70 μm, respectively. We numerically differentiate the incident field amplitudes. We assume that the incident field amplitude has a uniform spatial phase distribution and the numerical aperture of the imaging system is 0.02. The theoretical differentiation calculations are presented as the dashed lines in Fig. 2d. As shown in Fig. 2d, there is a very good agreement in the peak position between the experimental measurements and the numerical results of the spatial differentiation. Moreover, between the two peaks the

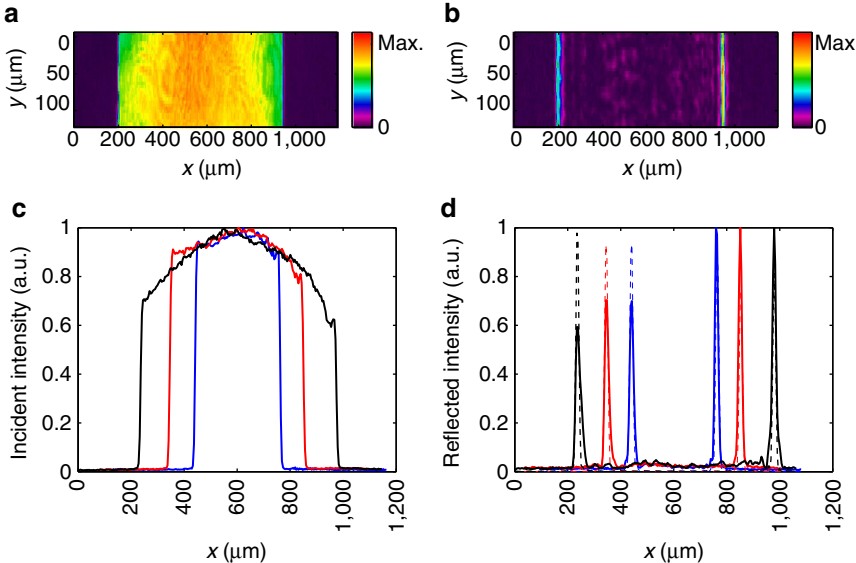

**Figure 2 | Spatial differentiation demonstration for the beams through slits.** (**a**) Incident field intensity generated by an adjustable slit. (**b**) Measured reflected intensity image corresponding to the incident field of **a**. (**c**) Normalized average intensity of the incident fields with three different slit widths of 321.65 μm (blue line), 504.20 μm (red line) and 738.70 μm (black line). (**d**) Experimental normalized average reflected intensity (solid lines) and theoretical differentiation results (dashed lines) for the three cases in **c**.

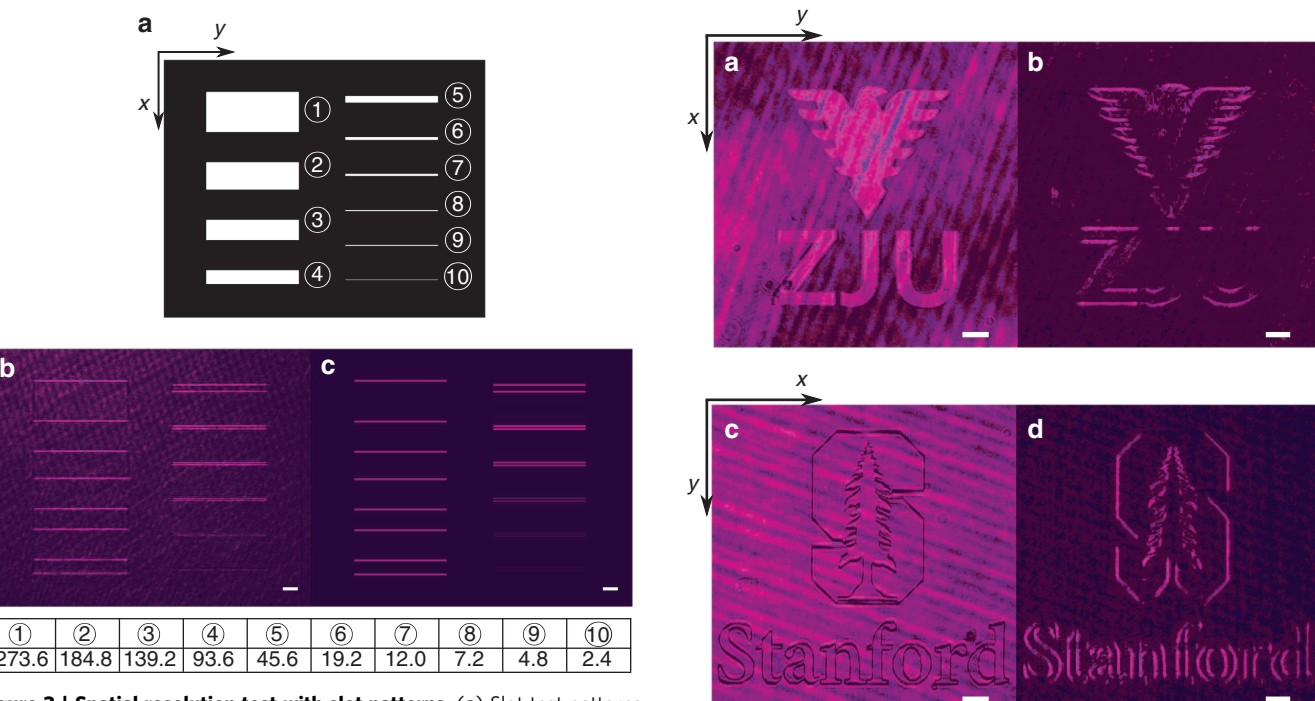

| ① | ② | ③ | ④ | ⑤ | ⑥ | ⑦ | ⑧ | ⑨ | ⑩ |
|------|------|------|------|------|------|------|------|------|------|
| 273.6 | 184.8 | 139.2 | 93.6 | 45.6 | 19.2 | 12.0 | 7.2 | 4.8 | 2.4 |

**Figure 3 | Spatial resolution test with slot patterns.** (**a**) Slot test patterns on the SLM with the different phases for the black and the white areas. (**b,c**) Reflected intensity image by experimental measurement (**b**) and numerical simulations (**c**). The white bars correspond to the length of 100 μm. Inset Table: the widths of the slots used in **a** in μm.

**Figure 4 | Edge detection for changes in either amplitude or phase.** (**a**) Incident image consisting of the ZJU eagle logo and letters with amplitude modulation, where the inside and the outside of the letters and the eagle have different intensities. (**b**) Reflected intensity image corresponding to **a**. (**c**) Incident image consisting of the Stanford tree logo and letters generated with phase modulation, where the inside and the outside of the letters and the logo have different phases but the same intensity. (**d**) Reflected intensity image corresponding to **c**. The white bars correspond to the length of 100 μm.

experimental measurement shows very low field intensity, again in agreement with the numerical differentiation results since the incident field has much slower spatial variation between the two edges. We calculate the Pearson's correlation coefficient between the reflected intensity and theoretical differentiation calculation. For the three cases, the correlation coefficients are 93.9%, 93.3% and 90.6%, respectively, which show the good performance of the plasmonic spatial differentiator.

**Edge detection.** We now demonstrate various aspects of our plasmonic differentiator that are important for high-throughput image processing. For this purpose, we generate various images

with a spatial light modulator (SLM: Holoeye PLUTO-NIR-011) and project the incident fields on the plasmonic differentiator with a conjugate imaging system (see Supplementary Note 2 and Supplementary Fig. 2). As the first illustration, since the spatial spectral transfer function of our plasmonic differentiator is linear only in the vicinity of $k_x = 0$ as shown in Fig. 1d, there is a finite spatial bandwidth of our plasmonic differentiator: The differentiator will not be able to resolve two edges that are very close to each other. As an experimental measurement of the spatial resolution of our plasmonic differentiator, we generate slot test patterns on the SLM (Fig. 3a), and the corresponding reflected intensity is measured and shown in Fig. 3b. As equation (3), the differentiation only operates along the $x$-direction. As a result, Fig. 3b only shows bright lines for the horizontal edges of each slot, corresponding to the $x$-direction differentiation. We also numerically simulate the reflected intensity as shown in Fig. 3c, which agrees well with the experimental results. We observe that the performance of the edge detection degrades as the width reduces. Figure 3b shows that the resolution of our plasmonic differentiator, that is the minimum separation between the two edges that can be resolved, is about 7.2 μm. Such a resolution should be sufficient for most image processing applications.

We note that in our device the differentiation operates on the field rather than the intensity. Therefore the device can be used to detect an edge either in the intensity or in the phase distribution of the incident field. To show such an effect, we use the SLM to generate incident fields with amplitude and phase modulations, respectively. Figure 4a shows an incident image field of the ZJU eagle logo and letters generated with amplitude modulation, where the inside and the outside of the letters and the eagle have different intensities. Figure 4b shows the reflected intensity from our plasmonic differentiator, which clearly exhibits the outlines of the logo and the letters. Figure 4c shows the incident image field consisting of the Stanford tree logo and letters generated with phase modulation, where the inside and the outside of the letters and the logo have different phases but the same intensity. We note that here we have rotated the incident image to demonstrate that we can detect the vertical edges. Again, the reflected light clearly exhibits the outline of the letters and logo (Fig. 4d). Since the differentiation is along the $x$-direction, the edges perpendicular to the $x$-direction are most visible. Nevertheless, as long as the edge is not along the $x$-direction, it can be detected in the reflected beam (e.g., the letter S in Fig. 4d).

## Discussion

We experimentally demonstrate that the single subwavelength metal film can compute spatial differentiation of the incident beam field during the light reflection. We exploit this feature to detect the edges that are formed by sharp variations of either the phase or the amplitude distributions in the incident field. The resolution of the edge detection is about 7 μm. Therefore, we have demonstrated an ultrafast optical computing scheme with sufficient space-bandwidth product that is capable of processing an entire image on a single shot, which is critically important for high-throughput real-time image processing. The structure used here can be fabricated at large scale with standard thin-film deposition technology. The thickness of the film is on the order of 50 nm, which tremendously miniaturizes the element as compared to lens-based edge detection schemes. Our results indicate that macroscopic-scale optical analog computing can be performed using simple nanoscale thin-film devices.

The plasmonic differentiator demonstrated here performs spatial differentiation along only in a single direction. Such a feature is already useful for image processing purposes for feature detection and classification[3]. As demonstrated in Fig. 4, one can simply perform differentiation along a perpendicular direction by rotating the beam. Alternatively, instead of the prism coupling method, by using a two-dimensional grating coupler that simultaneously excites two propagating SPP, one can construct plasmonic differentiators that simultaneously perform differentiation along two perpendicular directions.

Here, we control the interference by tuning the deposition thickness of the metal film. Fundamentally, with the technology of metamaterials, the interference process can be locally controlled through electrical, optical, thermal or magnetic approaches[37–39]. Therefore, the present plasmonic differentiator can be extended to a local dynamical spatial field converter with nanoscale modulation.

## Methods

**Sample fabrication and optical measurement.** A thin silver layer was deposited on a clean BK7 glass substrate using a magnetron sputtering system (Kurt J. Lesker) with a 40W radio frequency sputtering power. The substrate temperature was fixed at room temperature. The thickness of the silver film was monitored by a quartz oscillator during the deposition. To measure the spatial spectral transfer function in a Kretschmann configuration, the sample was pasted on the waist surface of a BK7 prism with the aid of the index matching oil. The prism was mounted on a rotating stage. A collimated green (532 nm) continuous-wave laser beam was expanded and illuminated on the metal surface with the polarization controlled by an adjustable polarizer. By rotating the prism to scan the incident angle, the intensities of the incident and reflected light were respectively measured by two silicon photo-diode powermeters (Thorlabs Inc.). The spatial spectral transfer function was obtained by normalizing the data. We used a beam profiler (Ophir SP620) to measure the incident and the reflected field profiles. We note that here the beam profiler is used instead of conventional charge-coupled devices because the charge-coupled device uses gamma correction and cannot accurately measure the field intensity.

**Data availability.** The authors declare that the data supporting the findings of this study are available within the paper and its Supplementary Information files.

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

## Acknowledgements

This work was supported by the Thousand Youth Talents Plan, Fundamental Research Funds for the Central Universities (2014QNA3007) and the National Natural Science Foundation of China (NSFC 61675179). H.Y. was supported by the National Basic Research Program of China (2013CB632104) and the NSFC (61575176). M.Q. was supported by the NSFC (61425023). S.F. was supported by the U.S. Air Force Office of Scientific Research (FA9550-12-1-0024 and FA9550-17-1-0002).

## Author contributions

Z.R. conceived the idea of plasmonic spatial differentiator and designed the experiment. T.Z. and Y.Z. built the experimental set-up and carried out the measurement. Z.R., T.Z., Y.L. and M.Q. carried out data analysis and performed simulations. H.Y. prepared the experimental samples. Z.R., S.F. and T.Z. co-wrote the paper. Z.R. and S.F. supervised and coordinated all the work.

## Additional information

Competing interests: The authors declare no competing financial interests.

