## [Peer Review File · Nature Communications]

Reviewers' comments:

Reviewer #1 (Remarks to the Author):

This paper presents a simple and efficient method for spatial differentiation. The differentiation is performed by a thin Ag layer illuminated by an obliquely incident wave in the Kretschmann configuration. The incident angle and the thickness of Ag layer is properly designed to achieve critical coupling of the incident wave to the SPP wave supported by the metal-dielectric interface. The performance of the device has been described by coupled mode theory and it has been demonstrated experimentally. This work is novel and very important for the field of optical analog computing. The paper can be accepted for publication in Nature Communications, provided that the following issues are addressed:

1- The authors correctly claim that their device is based on "the simplest surface plasmonic structure", however, a recent work (Youssefi, Amir, et al. "Analog computing by Brewster effect." Optics Letters 41.15 (2016): 3467-3470.) propose simpler structure for differentiation that is not mentioned by the authors.

2- The authors emphasize on the simplicity of their structure, although this is true but in my opinion the main superiority of their device is its high gain ($1/0.0075 = 133$) of differentiation. This high gain is because of working near the pole of the system. If the device did not have such a high gain of differentiation, even a small nonzero reflection at $k_x = 0$ (which is inevitable in practice and it can be seen in Fig. 1) would deteriorate the differentiator. This point should be highlighted in the paper.

3- As mentioned in the previous comment the transfer function has a pole near the desired zero. Although this pole leads to high gain but it limits the bandwidth of the device. As a result only wide beams which have narrow spatial bandwidth are successfully differentiated. This is the reason that the edges with separation smaller than 7.2 micron ($\sim 14\lambda$) cannot be resolved. This trade-off and its effect on the device performance should be explained in the paper.

Reviewer #2 (Remarks to the Author):

The submitted paper presents a novel and effective plasmonic platform to realize optical spatial differentiation. Using the proposed approach, the authors experimentally demonstrate optical edge detection of macroscopic images. Overall, the paper is well written and convincing, and the presented results are exciting as they demonstrate a novel practical route to perform optical analog computing without Fourier-optics setups. I'm sure the present paper will be of broad interest to the optics and metamaterials communities, and will inspire further investigations and practical designs for optical analog computing in the spatial domain. For these reasons, I support the publication of the submitted paper in Nature Communications, after the following minor points have been addressed:

- The authors write: "With the concept of metamaterial, Silva et al. theoretically showed that through a complex array of meta-atoms both phase and magnitude of the transmitted wave can be engineered [...] In contrast to the metamaterial approach for spatial differentiation, which relies on a complex array of meta-atoms [18, 20], the proposed plasmonic differentiator consists of a single ultra-thin metal film, which is far simpler and greatly simplifies fabrication process and reduces the influence of fabrication imperfection." While the first approach of Silva et al. relies indeed on a complex array of meta-atoms, and it is still analogous to a Fourier-optics setup, in the second approach they don't use an array of meta-atoms, but a multi-layered slab that is typically only a wavelength thick. The plasmonic-slab approach discussed in the present paper can be considered a smart simplification of the second approach of Silva et al.

- For $\theta_0 = 0$ (normal incidence), the transfer function of the system is an even function of k_x . Therefore, first-order spatial differentiation should not be possible (while even-order differentiation is possible). Since this fact is not directly inferred from Eqs. (1-3), the authors should stress that these formulas apply only for oblique incidence.

- It would be useful if the authors could quantify the frequency bandwidth over which the structure works as a spatial differentiator. Is it very narrow, or acceptable performance are obtained over a finite bandwidth?

- Typos: "possessing" instead of "processing", "differetiator".

Point-by-Point Response to Referee #1's Comments:

“This paper presents a simple and efficient method for spatial differentiation. The differentiation is performed by a thin Ag layer illuminated by an obliquely incident wave in the Kretschmann configuration. The incident angle and the thickness of Ag layer is properly designed to achieve critical coupling of the incident wave to the SPP wave supported by the metal-dielectric interface. The performance of the device has been described by coupled mode theory and it has been demonstrated experimentally. This work is novel and very important for the field of optical analog computing. The paper can be accepted for publication in Nature Communications, provided that the following issues are addressed: ”

Response: We thank the reviewer for the favorable high-level evaluation of this work, and for many insightful comments. Below we provide a point-by-point response to the reviewer's comments.

To Comment (1): *“The authors correctly claim that their device is based on "the simplest surface plasmonic structure", however, a recent work (Youssefi, Amir, et al. "Analog computing by Brewster effect." Optics Letters 41.15 (2016): 3467-3470.) propose simpler structure for differentiation that is not mentioned by the authors.”*

Response and modification: We thank the referee for pointing out this reference. We have modified the introduction and expanded the citation which included Ref.[28]:

(the 34th line in Page 1) “Recently, there are significant efforts seeking to miniaturize such computing elements down to a single-wavelength or even sub-wavelength scale [19-29]. ”

(the 44th line in Page 1) “There are also several other theoretical proposals, including one discussed by Silva et al.[19], where one seeks instead to achieve spatial differentiation with layer structures [26-28]. None of these structures, however, have been demonstrated experimentally either.”

[28] A. Youssefi, F. Zangeneh-Nejad, S. Abdollahramezani, and A. Khavasi, “Analog computing by Brewster effect,” *Optics Letters* 41, 3467–3470 (2016).

To Comment (2) and (3): *“The authors emphasize on the simplicity of their structure, although this is true but in my opinion the main superiority of their device is its high gain ($1/0.0075 = 133$) of differentiation. This high gain is because of working near the pole of the system. If the device did not have such a high gain of differentiation, even a small nonzero reflection at $kx = 0$ (which is inevitable in practice and it can be seen in Fig. 1) would deteriorate the differentiator. This point should be highlighted in the paper.”*

“As mentioned in the previous comment the transfer function has a pole near the desired zero. Although this pole leads to high gain but it limits the bandwidth of the device. As a result only wide beams which have narrow spatial bandwidth are successfully differentiated. This is the reason that the edges with separation smaller than 7.2 micron ($\sim 14\lambda$) cannot be resolved. This trade-off and its effect on the device performance should be explained in the paper.”

Response: We appreciate the referee's insightful comments. The relatively small value

of the parameter B in Eq.(3) indeed enhances the throughput of the spatial differentiation, which is of great benefit to the experiments. But it also limits the spatial bandwidth of the differentiation.

Modification: To show this trade-off, we have added the following discussion:

(the 181th line in Page 2) “We note that the relatively small value of B enhances the output signal obtained from the spatial differentiation (c.f. Eq.(3)). On the other hand, such a small value also means that the linear dependency is preserved only near $k_x = 0$ within a narrow spatial bandwidth (Fig. 1d).”

Point-by-Point Response to Referee #2's Comments:

“The submitted paper presents a novel and effective plasmonic platform to realize optical spatial differentiation. Using the proposed approach, the authors experimentally demonstrate optical edge detection of macroscopic images. Overall, the paper is well written and convincing, and the presented results are exciting as they demonstrate a novel practical route to perform optical analog computing without Fourier-optics setups. I’m sure the present paper will be of broad interest to the optics and metamaterials communities, and will inspire further investigations and practical designs for optical analog computing in the spatial domain. For these reasons, I support the publication of the submitted paper in Nature Communications, after the following minor points have been addressed: ”

Response: We thank the reviewer for the favorable evaluation of the technical quality of the work and for many insightful comments. Below we provide a point-by-point response to the reviewer’s comments.

To Comment (1): *“While the first approach of Silva et al. relies indeed on a complex array of meta-atoms, and it is still analogous to a Fourier-optics setup, in the second approach they don’t use an array of meta-atoms, but a multi-layered slab that is typically only a wavelength thick. The plasmonic-slab approach discussed in the present paper can be considered a smart simplification of the second approach of Silva et al.”*

Response and Modification: We agree. We have modified the introduction and explicitly mentioned the multilayer approach in Silva’s work.

(the 34th line in Page 1) “Recently, there are significant efforts seeking to miniaturize such computing elements down to a single-wavelength or even sub-wavelength scale [19-29]. ”

(the 44th line in Page 1) “There are also several other theoretical proposals, including one discussed by Silva et al.[19], where one seeks instead to achieve spatial differentiation with layer structures [26-28]. None of these structures, however, have been demonstrated experimentally either.”

To Comment (2): *“For $\theta_0 = 0$ (normal incidence), the transfer function of the system is an even function of k_x . Therefore, first-order spatial differentiation should not be possible (while even-order differentiation is possible). Since this fact is not directly inferred from Eqs. (1-3), the authors should stress that these formulas apply only for oblique incidence.”*

Response and Modification: We thank the referee for the suggestion to the symmetry discussion. To clearly show this point we have added the following sentence:

(the 128th line in Page 2) As a few remarks, we note that here we are able to demonstrate the first-order derivative since the plasmonic differentiator operates away from the normal incidence. This is in contrast to Ref. [19], where with a multilayer film a second-order derivative is demonstrated for normal incidence due to the symmetry constraint on both the flat structure and the light source.

To Comment (3): “It would be useful if the authors could quantify the frequency bandwidth over which the structure works as a spatial differentiator. Is it very narrow, or acceptable performance are obtained over a finite bandwidth?”

Response and Modification: We greatly appreciate the referee’s insightful suggestion. Currently the frequency bandwidth of the plasmonic differentiator cannot be measured with the present experiment setup. To estimate the frequency bandwidth, we consider the case that the incident light is a plane wave in the space-domain but with a pulse shape in the time domain. We numerically simulate the frequency-domain transfer function and the results are shown in Fig. R1. As expected, if the incident field bandwidth is relatively narrow at the operating frequency 563.9THz (corresponding to the wavelength of $0.532 \mu m$), the throughput is low which coincides with the spatial differentiation to the plane wave. Moreover, Fig. R1 shows that due to the low Q-factor of the system the transfer function remains below -10dB over a range with a bandwidth of 13 THz. Therefore, it also indicates that the plasmonic differentiator has enough large frequency bandwidth for the ultrafast image processing.

Fig. R1 The frequency-domain transfer function of the plasmonic differentiator.

We have added a few sentences in the manuscript to discuss the frequency bandwidth:

(the 134th line in Page 2) Also, with a relatively low quality factor of the plasmonic resonance, when maintaining the incident angle, the reflection remains low with a THz bandwidth at the resonant frequency. Therefore, the spatial differentiator proposed here should have enough frequency bandwidth for ultrafast image processing.

To Comment (4): *Typos: “possessing” instead of “processing”, “differetiator”.*”

Response and Modification: We thank the referee’s notification and have corrected these typos and done more proofreading to the manuscript.

REVIEWERS' COMMENTS:

Reviewer #1 (Remarks to the Author):

I recommend this paper for publication.

Reviewer #2 (Remarks to the Author):

The authors have properly addressed all the questions and comments raised by the reviewers. I think the revised paper is ready for publication in Nature Communications.